# Nitrogen Fixation Activity and Genome Analysis of a Moderately Haloalkaliphilic Anoxygenic Phototrophic Bacterium *Rhodovulum tesquicola*

**DOI:** 10.3390/microorganisms10081615

**Published:** 2022-08-09

**Authors:** Anastasia V. Komova, Elizaveta D. Bakhmutova, Anna O. Izotova, Evelina S. Kochetova, Stepan V. Toshchakov, Zorigto B. Namsaraev, Maxim V. Golichenkov, Aleksei A. Korzhenkov

**Affiliations:** 1Kurchatov Centre for Genome Research, NRC Kurchatov Institute, 123098 Moscow, Russia; 2Soil Biology Department, Soil Science Faculty, Moscow State University, 119991 Moscow, Russia; 3Department of Biology, Faculty of Pediatrics, Pirogov Russian National Research Medical University, 117997 Moscow, Russia

**Keywords:** anoxygenic phototrophs, halophiles, alkaliphiles, nitrogen fixation, diazotroph, *Rhodovulum tesquicola*, genome sequence

## Abstract

The genome of the moderately haloalkaliphilic diazotrophic anoxygenic phototrophic bacterium *Rhodovulum tesquicola* A-36s^T^ isolated from an alkaline lake was analyzed and compared to the genomes of the closest species *Rhodovulum steppense* A-20s^T^ and *Rhodovulum strictum* DSM 11289^T^. The genomic features of three organisms are quite similar, reflecting their ecological and physiological role of facultative photoheterotrophs. Nevertheless, the nitrogenase activity of the pure cultures of the studied bacteria differed significantly: the highest rate (4066 nmoles C_2_H_2_/mg of dry weight per hour) was demonstrated by *Rhodovulum strictum* while the rates in *Rhodovulum tesquicola* and *Rhodovulum steppense* were an order of magnitude lower (278 and 523 nmoles C_2_H_2_/mg of dry weight per hour, respectively). This difference can be attributed to the presence of an additional nitrogenase operon found exclusively in *R. strictum* and to the structural variation in nitrogenase operon in *R. tesquicola*.

## 1. Introduction

The genus *Rhodovulum* includes anoxygenic phototrophic bacteria that require sodium chloride for growth [1,2]. The members of the genus are widely distributed in marine and saline environments [3,4,5], and due to their metabolic capacities may be used in a variety of biotechnological applications [6,7,8]. Additionally, recent studies show a wider spread of photoferrotrophy and phototrophic extracellular electron uptake in *Rhodovulum* that previously thought [9]. Thus, the genome analysis of the representatives of the genus appears to provide new insights into the genetic and evolutionary basis of metabolic processes in halotolerant anoxygenic phototrophic *Proteobacteria*. To date, *Rhodovulum* comprises 20 validly described species, for eight of which no genome sequence has been published so far.

Bacteria of the genus *Rhodovulum* preferably grow photoheterotrophically and are able to fix nitrogen. Thus, they serve as an important mediator of the carbon and nitrogen cycles as it links the organic matter decomposition and nitrogen fixation. Saline and soda lakes are among the most productive ecosystems in the world [10,11]. The alkaline conditions favour the volatilization of ammonium from the lakes; therefore the regular supply of nitrogen is needed to provide the functioning of ecosystem [12].

The process of nitrogen fixation in saline and soda lakes was studied mostly for the natural ecosystems and mesocosms [13,14,15,16,17]. The nitrogen fixation in pure cultures of phototrophic *Alphaproteobacteria* was studied in details for the neutrophilic and some of the marine and true halophilic representatives [18,19,20]. Nevertheless, little is known about this process in pure cultures of halotolerant and alkaliphilic anoxygenic phototrophic *Alphaproteobacteria* that are widespread in the lakes with relatively low salinity and increased alkalinity [5,21].

*Rhodovulum tesquicola* A-36s^T^ was isolated from an alkaline lake Sul’fatnoe (Buryat Republic, Siberia, Russia; pH 9.2, mineralization 7.7 g/L) [5] and later described as a type strain of a new species within *Rhodovulum* genus [22]. It is a typical anoxygenic phototrophic bacterium able to grow photoheterotrophically and chemoheterotrophically with a range of organic substrates, and to grow photolithoautotrophically with sulfur compounds as electron donors. However, its ability to fix nitrogen has not been studied before. Here we present the analysis of genome of *R. tesquicola* in comparison with the genomes of the closest species, *R. steppense* A-20s^T^ and *R. strictum* DSM 11289^T^, and the data on the nitrogen fixation activity by the pure cultures of these organisms.

## 2. Materials and Methods

### 2.1. Microbial Cultivation

*R. tesquicola* A-36s^T^ (=ATCC BAA-1573^T^), *R. steppense* A-20s^T^ (=DSM 21153^T^) (both isolated previously by A.V.K. and kept at the laboratory collection) and *R. strictum* DSM 11289^T^ obtained from the Leibniz-Institut DSMZ, Braunschweig, Germany were grown photoheterotrophically at 30 °C on the previously described medium [22] (pH 8.5–9).

### 2.2. Nitrogenase Activity Measurement

Nitrogenase activity measurement was performed by the acetylene reduction assay [23]. 1 mL of acetylene (about 10% of gaseous headspace in the vial) was injected into the hermetically sealed 23 mL vials containing liquid photoheterotrophically grown cultures of the investigated bacteria with argon filled headspace. The vials were incubated at 25 °C for 20 h. After that, 1 mL of gaseous phase was sampled and the detection of ethylene was performed on Crystal-2000 chromatograph (LLC NPF Meta-Chrom, Yoshkar-Ola, Russia) with a heated flame ionization detector (HFID). Instrument characteristics: column length—1 m, diameter—3 mm, filler—Porapak N 80/100, column temperature—60 °C, detector temperature—160 °C, evaporator temperature—100 °C, carrier gas flow (N_2_)—50 mL/min, air—280 mL/min, hydrogen—28 mL/min.

### 2.3. Genome Sequencing and Assembly

Genomic DNA was isolated from cell culture of *R. tesquicola* A-36s^T^ using the QIAamp^®^ DNA Mini kit following the manufacturer’s recommendations (Qiagen, Dusseldorf, Germany).

DNA was fragmented by ultrasound on a Covaris M220 (Covaris, Woburn, MA, USA). A paired-end genomic library (average insertion size 300 bp) was made using the NEBNext^®^ Ultra^TM^ II DNA Library Prep Kit (New England Biolabs, Ipswich, MA, USA). The DNA library was sequenced using an Illumina NovaSeq 6000 System (Illumina, San Diego, CA, USA) with a 2 × 250 bp paired-end read.

Genome assembly was performed using ZGA software pipeline v0.1 [24]. Sequencing reads were processed using BBTools software toolkit v38.96 [25]. Reads with low quality bases and adapter sequences were trimmed from sequencing reads with BBduk v38.96 (ktrim = r qtrim = rl trimq = 21 k = 20 mL = 33 trimpolyg = 4 trimpolya = 4). Trimmed reads were corrected with Tadpole (mode = correct, cecc = t). Corrected reads were normalized with BBnorm v38.96 to 250× depth coverage. Normalized reads were assembled to scaffolds with SPAdes v3.14 [26] using default parameters. The resulting assembly was polished with Pilon v1.24 [27] using trimmed reads.

Primary annotation was performed using DFAST v1.2.15 [28], CDS were predicted with Prodigal. Additional annotation was performed with NCBI Prokaryotic Genome Annotation Pipeline (PGAP) v6.1 [29], proteins were predicted using GeneMarkS-2+ [30] and best-placed reference protein set. KEGG orthologs (KO) annotation was performed using KofamKOALA [31]. Completeness for a given KEGG module [32] was calculated using the program anvi-estimate-metabolism from anvi’o toolkit v7.1 [33].

All available on the 1 July 2022 *Rhodovulum* genomes including the genomes of *R. steppense* A-20s^T^ and *R. strictum* DSM 11289^T^ were downloaded from NCBI Assembly database. Genome completeness was assessed with CheckM v1.2.0 [34] taxonomy workflow using *Rhodobacteraceae* marker set. Pairwise average nucleotide identity (ANI) was estimated using FastANI [35]. Pangenomic analysis was conducted using Proteinortho v6 [36] with BLASTP+ [37] algorithm. Nitrogen fixation genes in *Rhodovulum* genomes were searched using gene annotations in NCBI protein and Uniprot and BLASTP+ [37] search in NCBI nr and Swissprot databases. Phi29-like polymerase and viral single-stranded DNA-binding protein were searched using BLASTP+ [37] (e-value 0.001) with P03680 and Q38504 reference sequences, respectively.

Amino acid sequences of nifH gene homologs were found via BLASTP+ [37] search in NBCI nr database. The first 250 hits were downloaded and clustered using CD-HIT [38] with 95% identity threshold. Representative sequences of 16S rRNA and *nifH* genes were aligned using MAFFT v7.475 with the FFT-NS-i algorithm [39]. The resulting alignments were manually curated and phylogenetic trees were inferred using FastTree v2.1.10 [40] in auto mode. The trees were visualized using ITOL web-server [41].

## 3. Results and Discussion

### 3.1. General Characteristics of Genome

The sequencing yielded in 6,272,843 pair-end reads with a length of 251 bp, totaling to 41.39 Gbp. The draft genome of *R. tesquicola* A-36s^T^ consists of 80 sequences resulting in 3,521,822 bp with GC-content of 67.68% and N50 of 156,513 bp (Table 1). The most similar to *R. tesquicola* A-36s^T^ genomes of cultivated *Rhodovulum* species were *R. steppense*—ANI value 94.86% and *R. strictum*—88.91% (Figure 1 and Figure 2). These values support the novelty of *R. tesquicola*.

The completeness of the KEGG modules was assessed for *R. tesquicola*, *R. steppense* and *R. strictum* genomes. The completeness differs only in 28 out of 204 modules. Only 18 differing modules are complete for at least 50% for at least one of the genomes and differ for at least 10% (Figure 3, Appendix A).

Pangenome analaysis of three *Rhodovulum* genomes (Figure 4) showed that *R. tesquicola* has the smallest genome and more common protein clusters comparing to *R. steppense* and *R. strictum*. *R. tesquicola* and *R. steppense* have more in common than any other pair of genomes that emphasize their proximity. The core genome of the three *Rhodovulum* genomes consists of 2646 protein clusters corresponding to 79% of *R. tesquicola* genome. Among genes lacking in *R. tesquicola* genome there are mobile elements, genes of prophage loci and hypothetical proteins without any functional annotation. Phi29-like DNA polymerase and phage single-stranded DNA-binding protein genes were not found in any of the three genomes.

### 3.2. Carbon Metabolism

Genes responsible for a complete TCA cycle, non-oxidative phase of pentose phosphate pathway, Entner-Doudoroff pathway, glycolysis and gluconeogenesis are present in all the genomes studied (Appendix A). The anoxygenic phototrophic bacteria can assimilate acetate as a sole organic substrate in a number of anaplerotic pathways to replenish the oxaloacetate pool [42,43]. Among them, the genes for the full ethylmalonyl pathway were found in the genome of *R. tesquicola*. The methylaspartate and glyoxylate pathways are incomplete. Although there is evidence for the isocitrate lyase presence in *R. steppense* [44], we have found no corresponding or homologous genes in any of the three genomes studied. Thus, the detection of the potential presence and functioning of glyoxylate cycle is a subject of further research. The absence of gene for acetyl-CoA synthetase in *R. tesquicola* shows that only the two-step mechanism of acetate assimilation via acetyl phosphate may function in *R. tesquicola*, while *R. steppense* and *R. strictum* have genes for both ways [45]. On the opposite, the gene for malonyl-CoA decarboxylase potentially involved in malonyl/methylmalonyl utilization [46,47] was detected only in *R. tesquicola*, thus expanding the variety of organic substrates for utilization.

### 3.3. Nitrogen Metabolism

During photoheterotrophic growth, the nitrogenase activity of *R. strictum* was significantly higher than that of *R. steppense* and *R. tesquicola* (Table 2 and Table 3).

The genomes of all three investigated bacteria contain the genes necessary for the biosynthesis of Mo-nitrogenase (*nifHDK*) and iron–molybdenum cofactor FeMoco (*nifB*, *nifEN*, *nifV*); no alternative nitrogenases were found.

All three genomes share a common nitrogenase operon (“slow” operon) (Figure 5). These genes (GH815_10410-GH815_10390 in case of *R. strictum*) have close homologues in several representatives of the genera *Rhodobacter* and *Cereibacter*, as well as in a small number of *Alphaprotobacteria* mainly from *Hyphomicrobiales*. Relatively close homologues are also found in the representatives of *Firmicutes* and *Chloroflexi* (Appendix A), suggesting a lateral gene transfer event in case of the “slow” operon. In addition, we have found the second nitrogenase operon in *R. strictum* genome (Figure 5). Only a limited number of *Alphaproteobacteria* including *R. strictum* and *Rhodobacter capsulatus* possesses second operon with putative higher nitrogen fixation activity (“fast” operon) (Figure 6).

The *nif* operon in *R. tesquicola* is broken into two parts. 16 genes from *nifA* (NHN26_05215) to *fdxB* (NHN26_05140) are separated from *nifU* (NHN26_05060), *nifV* (NHN26_05055) and *nifW* (NHN26_05050) by a cluster of 15 genes coding mainly for *RnfABCDGE* type electron transport complex. This break may lead to a decreased nitrogenase activity in comparison with *R. steppense*.

Nitrogen fixing phototrophic microorganisms play an important role in the nitrogen uptake by the phototrophic communities of the saline and soda lakes. It has been previously demonstrated that in soda lakes, nitrogen is fixed mainly at low salinities with the nitrogen fixation rates in the light much higher than those in the dark at the salinities of less than 100 g/L [17,48]. The investigated communities consisted mostly of the photoautotrophic heterocystous cyanobacteria with *nifH*-containing *Ectothiorhodospira* sp. present to a lesser extent. In the saline and soda lakes, *Ectothiorhodospiraceae* usually occur at the salinities up to the saturation, while *Rhodobacteraceae* reach high numbers at the mineralization below 100 g/L [21], and thus may be responsible for the nitrogen fixation activity at these conditions. However, in spite of wide distribution of *nif*-related genes among the phototrophic *Alphaproteobacteria* inhabiting saline and soda environments [49], the actual rates of nitrogen fixation in pure cultures as well as *in situ* may vary strongly (Table 3).

The product of the *nifH* gene is dinitrogenase reductase, also known as the Fe protein or component II. The Fe protein has an obligate redox role and is involved in the process of maturation of the P-clusters of the apo-MoFe protein to their catalytically active forms [50,51,52]. Probably the “fast” and “slow” operon function differs in the expression rate of *nifH* genes, leading to the different rates of nitrogen fixation. As *R. strictum* and *Rba*. *capsulatus* contain 2 operons with different *nifH* variants (Figure 5) while *R. steppense* and *R. tesquicola* have a single operon, we hypothesize that either the “fast” operon or both of them are responsible for high nitrogenase activity. Phylogenetically, the “fast” operon cluster (Appendix A) includes not only *Rba. capsulatus* and *R. strictum* performing high nitrogenase activity but also *Rhodoblastus sphagnicola*, *Rhodomicrobium vannielii* and *Rhodopseudomonas palustris* with moderate nitrogenase activity, so it is likely that both “fast” and “slow” operons may be required to achieve high activities.

The structure and organization of genes for molybdenum-only and alternative nitrogenases in *Rba. capsulatus* and other organisms are well studied [50,52,53,54,55]. The presence of multiple copies of *nif* genes in *Rba. capsulatus* was demonstrated [56], but little is known about the nitrogenase activity depending on the number and structure of nitrogenase operon. The phylogenetic studies of nitrogenase on the phyla level show that most probably nitrogenase first appeared in the methanogenic archaea and later *nif* genes entered bacteria via horizontal gene transfer [57,58]. Nevertheless, the taxa within phylum can have different nitrogenases that can function in a different way depending on the environmental conditions and physiological features [58]. For example, it was demonstrated that *Anabaena variabilis* possesses two types of nitrogenase that function under different environmental conditions and express in different types of cells [59]. In our study, we show that two types of nitrogenase operons are present in the *Rhodovulum* species that probably results in different nitrogen fixation rates, so it is necessary to further investigate this phenomenon.

Unlike *R. tesquicola* and *R. steppense*, both isolated from saline soda lakes, *R. strictum* was isolated from a coastal colored bloom [60]. In spite of the fact that purple nonsulfur bacteria do not predominate in the coastal colored blooms, the members of *Rhodovulum* may represent a significant proportion of the phototrophic bacterial population in such environments [3]. It is probably due to the high nitrogenase activity that gives them the competitive advantage in case of nitrogen removal during the bloom.

### 3.4. Vitamin Requirements

The vitamin requirements of the investigated bacteria according to the physiological tests are given in Table 2. Biotin requirement corresponds with the fact that biotin biosynthesis pathways are incomplete in the genome of all three bacteria. In contrast to *R. steppense* and *R. strictum*, *R. tesquicola* contains a gene for phosphomethylpyrimidine synthase which may allow it to start the biosynthesis of thiamine monophosphate. The physiological tests, however, demonstrate that *R. tesquicola* requires additional thiamine. Some of the genes involved into thyamine synthesis are probably not expressed for reasons yet to be discovered. The niacin requirement in *R. steppense* and *R. tesquicola* may be due to the absence of *nadAB* genes which in turn are present in *R. strictum*.

### 3.5. Osmotic Adaptation

*R. tesquicola* and *R. steppense* are more halotolerant than *R. strictum* (Table 2). Apart from the complete pathway of ectoin biosynthesis and a partial trehalose synthesis pathway in genomes of the former two bacteria (Figure 3), the ectoine/hydroxyectoine TRAP transporter genes *TeaABC* [61] and glycine betaine uptake system *proVWX* [62] were found. The distribution of the compatible solutes biosynthesis and uptake genes in the investigated moderately halophilic bacteria correlates with the previously described pattern [63].

## 4. Conclusions

The genomes of *R. tesquicola*, *R. steppense* and *R. strictum* demonstrate significant similarity, except for some features including the vitamin synthesis pathways and the repertoire of the nitrogen fixation genes. The analysis of genome and nitrogen fixation rates shows that the studied *Rhodovulum* members are capable of fixing nitrogen with different rates, and thus may play an important role in nitrogen fixation in high-salt and alkaline conditions. Nevertheless, it becomes evident that the presence of representatives of this group does not necessarily indicate the effective nitrogen fixation in these ecosystems. Moreover, the presence of the “fast” operon does not predict high rates of nitrogen fixation in an organism. Literature data and our results show that the relationship between nitrogen fixation rate and the nitrogenase operon structure is still poorly studied, and thus further studies in this direction appear promising.

## Figures and Tables

**Figure 1 microorganisms-10-01615-f001:**
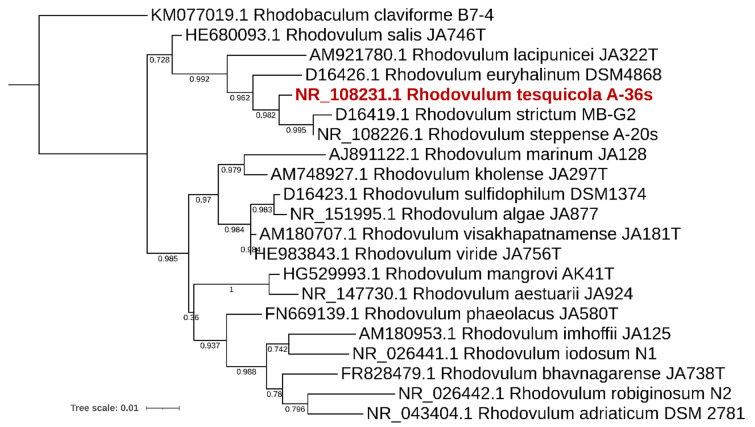
16S rRNA gene phylogeny of the genus *Rhodovulum*. The tree is scaled by evolutionary distance (count of substitutions per site).

**Figure 2 microorganisms-10-01615-f002:**
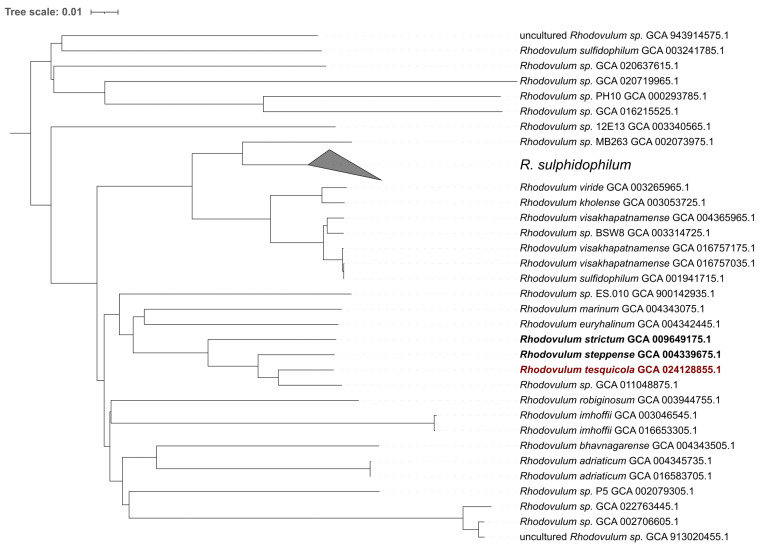
ANI dissimilarity dendrogram of high quality public *Rhodovulum* genomes. The dendrogram is rooted at the midpoint and scaled by genomic distance between strains.

**Figure 3 microorganisms-10-01615-f003:**
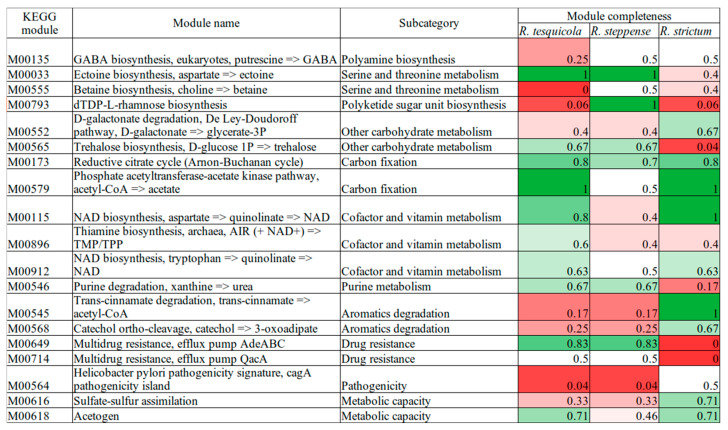
Completeness of the most different KEGG modules for three *Rhodovulum* genomes. Colors indicate the completeness of the module: red for the least complete, green for the most complete.

**Figure 4 microorganisms-10-01615-f004:**
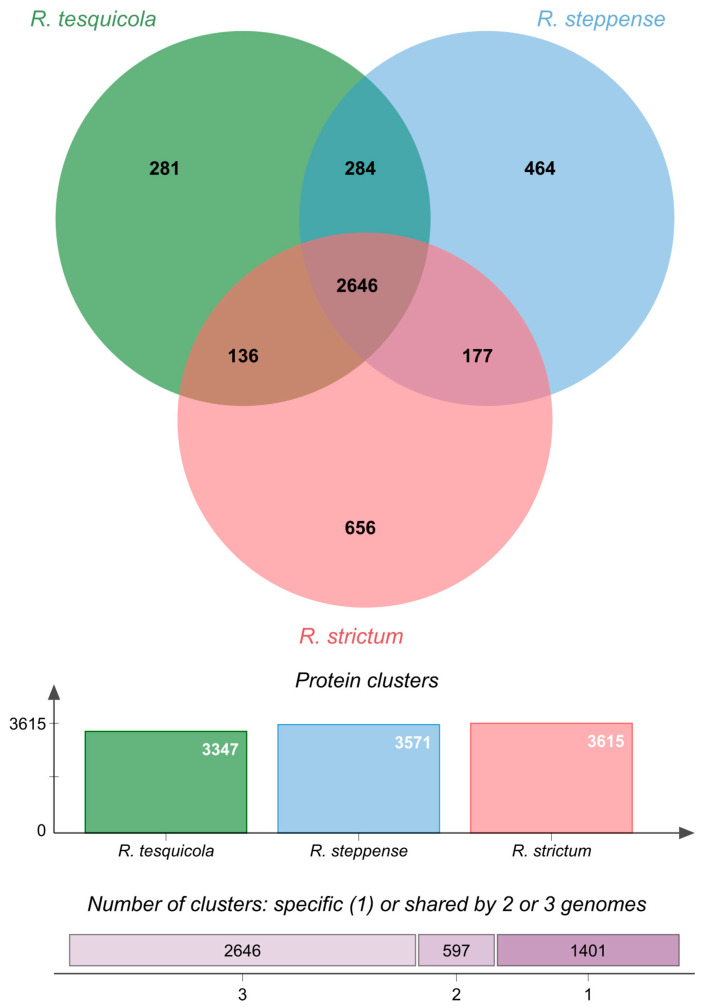
Venn diagram summarizing pangenome analysis of *R. tesquicola*, *R. steppense* and *R. strictum*.

**Figure 5 microorganisms-10-01615-f005:**
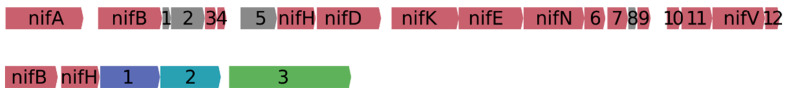
Nitrogen fixation operons. Top: common nitrogen fixation operon of three studied *Rhodovulum* species (“slow” operon): 1—4Fe-4S binding protein, 2—leucine rich repeat (LRR) protein, 3—*nifZ*, 4—*nifT*, 5—SIR2-like protein, 6—*nifX*, 7—*nifX*-associated nitrogen fixation protein, 8—hypothetical protein, 9—*nif*-specific ferredoxin III, 10—iron-sulfur cluster assembly accessory protein, 11—*nifU*, 12—*nifW*. Bottom: additional operon for nitrogen fixation present in *R. strictum* (“fast” operon): 1—oxidoreductase, 2—nitrogenase, 3—molybdopterin-dependent oxidoreductase (internal stop codon).

**Figure 6 microorganisms-10-01615-f006:**
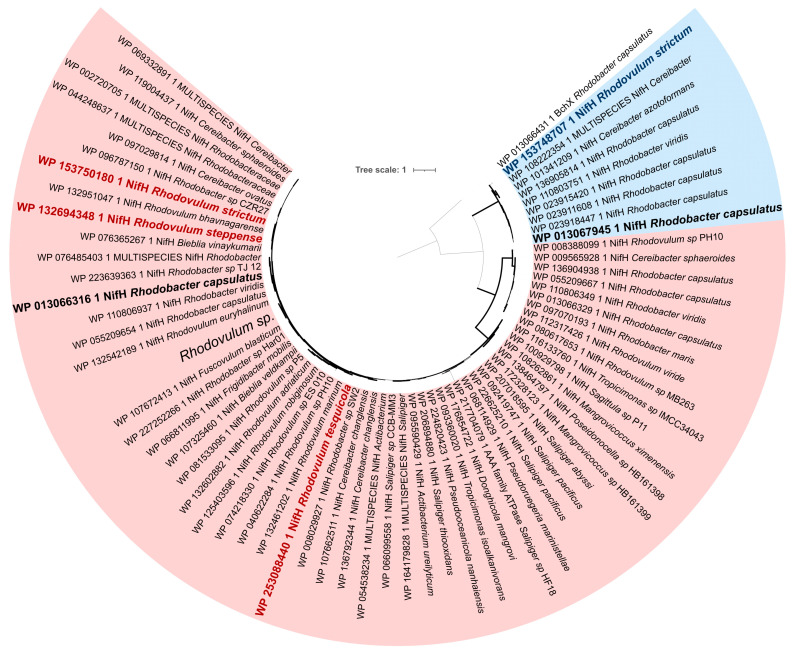
Phylogeny of *nifH* gene across *Rhodobacterales*. The tree includes *nifH* homologues missing in *R. steppense* and *R. tesquicola* but found in *R. strictum* and *Rhodobacter capsulatus* strain SB1003 (blue background) and *nifH* homologues appearing in all three *Rhodovulum* strains (red background). Ultrafast bootstrap values are displayed as branch width. The tree is scaled by evolutionary distance (count of substitutions per site).

**Table 1 microorganisms-10-01615-t001:** Genome features of *R. tesquicola*, *R. steppense* and *R. strictum*, according to NCBI database.

	*Rhodovulum tesquicola* A-36s^T^	*Rhodovulum steppense* A-20s^T^	*Rhodovulum strictum* DSM 11289^T^
Genome size	3,521,822	3,665,949	3,825,097
Total number of genes	3490	3634	3756
Proteins	3375	3582	3636
rRNA genes	1, 1, 1 (5S, 16S, 23S)	1, 1, 2 (5S, 16S, 23S partial)	1, 1, 1 (5S, 16S, 23S)
tRNA genes	45	51	42
Genome completeness, %	99.41	99.12	98.83
Genome contamination, %	0	0.15	0.59

**Table 2 microorganisms-10-01615-t002:** Physiological features of *R. tesquicola*, *R. steppense* and *R. strictum* ([22], this study). b, Biotin; paba, p-aminobenzoate; t, thiamine; n, niacin; +, high nitrogenase activity rate; +−, moderate nitrogenase activity rate.

	*R. tesquicola* A-36s^T^	*R. steppense* A-20s^T^	*R. strictum* DSM 11289^T^
pH range (optimum)	7.5–10.0 (8.5–9.0)	7.5–10.0 (8.5)	7.5–9.0 (8.0)
NaCl range (optimum) (%)	0.3–10.0 (1–3)	0.5–10.0 (1–5)	0.25–3.00 (0.8)
Vitamin requirement	b, paba, t, n	b, n, t	b, paba, t
Nitrogenase activity	+−	+−	+

**Table 3 microorganisms-10-01615-t003:** Nitrogenase activity in the pure cultures of anoxygenic phototrophic *Alphaproteobacteria* (strain designation is indicated in case if only one strain was tested).

Species	Nitrogenase Activity (nmoles C_2_H_2_/mg Dry Weight per Hour)	Reference
*Rhodobacter capsulatus*	1430–4900	[19]
*Rhodobacter capsulatus* B10	3650	[19]
*Cereibacter sphaeroides*	1110–2755	[19]
*Rhodovulum sulfidophilum* W4	640	[19]
*Rhodothalassium salexigens* DSM 2172	630	[19]
*Rhodomicrobium vannielii*	660–1570	[19]
*Rhodopseudomonas palustris*	480–700	[19]
*Rhodoblastus acidophilus*	500–900	[19]
*Rhodovulum tesquicola* A-36s^T^	278	This work
*Rhodovulum steppense* A-20s^T^	523	This work
*Rhodovulum strictum* DSM 11289^T^	4066	This work

## Data Availability

The *Rhodovulum tesquicola* A-36s^T^ sequencing reads were deposited in the NCBI database under BioProject PRJNA853128, genome submission ID SUB11687731.

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
