# Peer review of "Nitrogen Fixation Activity and Genome Analysis of a Moderately Haloalkaliphilic Anoxygenic Phototrophic Bacterium *Rhodovulum tesquicola"

_microorganisms, 2022, doi:10.3390/microorganisms10081615_

Round 1
Reviewer 1 Report
This manuscript describes nitrogen fixation activity and genome analysis of the moderately haloalkaliphilic phototrophic bacterium Rhodovulum tesquicola compared with two related species, Rhv. steppense and Rhv. strictum. The methodology and the approach to research are sound and the ms provides interesting information about genomes and nitrogen fixing activity of the phototrophs and its ecophysiological implications. I have only minor comments as follows:
1. Title. The term “diazotrophic” may be deleted.
2. L63. The sources of the test organisms should be given in more detail. The type strain should be with superscript “T”. What does the strain name A-20s (DSM 21153) mean?
3. L130-145. The strain names should be given behind the species names as appropriate.
4. Figs. 1 and 2. Specify the algorithm used for re-constructing the phylogenetic trees. Also what are the outgroups used in the trees? What are the figure “0.01” as tree scale? This is the case in Fig.6.
5. All gene names and scientific names should be in italic.
6. The author found the highest nitrogen fixation activity in Rhv. strictum, which was originally isolated from colored blooms developing in coastal area (IJSM 45, 319-326 [1995]). Concerning this, high nitrogen fixation activity of Rhv. strictum might explain why this organism exhibited massive growth in the colored blooms. The author could discuss in this respect.
Author Response
The authors thank the Reviewers for the valuable comments that helped a lot to improve the manuscript.
Reviewer 1.
- The term “diazotrophic” may be deleted.
The term “diazotrophic” is deleted from the title.
L63. The sources of the test organisms should be given in more detail. The type strain should be with superscript “T”. What does the strain name A-20s (DSM 21153) mean?
The sources of the strains are indicated.
The superscript “T” is added to the type strain names in Materials and Methods section. The strains` personal names as well as their numbers in the international collections are provided in Materials and Methods section.
- L130-145. The strain names should be given behind the species names as appropriate.
As the species names of the studied Rhodovulum strains do not coincide with other species mentioned in the text, the genus name “Rhodovulum” used with the species name was abbreviated to R. for convenience. Besides, as there are no other Rhodovulum strains different from those mentioned in the Materials and Methods section, we suppose that there is no need to indicate the strain name every time.
- Figs. 1 and 2. Specify the algorithm used for re-constructing the phylogenetic trees. Also what are the outgroups used in the trees? What are the figure “0.01” as tree scale? This is the case in Fig.6.
We have specified the algorithm in the Materials and Methods section. The information about the outgroups and evolutionary distances are now added to the Figures descriptions.
- All gene names and scientific names should be in italic.
Fixed.
- The author found the highest nitrogen fixation activity in Rhv. strictum, which was originally isolated from colored blooms developing in coastal area (IJSM 45, 319-326 [1995]). Concerning this, high nitrogen fixation activity of Rhv. strictum might explain why this organism exhibited massive growth in the colored blooms. The author could discuss in this respect.
The paragraph concerning the potential role of R. strictum is added to the “Results and Discussion” section.
Reviewer 2 Report
The manuscript requires English language editing. Both minor errors require correction, but larger grammatical issues prevent assessment of results and impact.
As one of the biggest points of interest for this work is the differential nitrogenase rates in the presented strain, more work needs to be done to clarify the nature of this feature. Deeper discussion of the published research on the two nitrogenase types, especially in R. capsulatus or others with two nitrogenase operons, would help support the proposition that there is a "fast" or "slow" nitrogenase. To facilitate this, Fig. 6 needs to be completely overhauled, as it is currently far too large with too small text and nothing can be interpreted regarding the nitrogenase operons. A simpler tree or trees showing the two Rdv. tesquicola clusters' relationship to other species is necessary to show this relationship.
Fig. 1 & 2 These are redundant and both are not necessary for the main body of the manuscript. I would recommend moving one to supplemental.
Fig. 1 & 6 Please clearly highlight the data sequenced in this work. The text is also incredibly small on both figures.
Author Response
The authors thank the Reviewers for the valuable comments that helped a lot to improve the manuscript.
- The manuscript requires English language editing. Both minor errors require correction, but larger grammatical issues prevent assessment of results and impact.
We have consulted a native English speaker and made corrections according to his/her comments.
- As one of the biggest points of interest for this work is the differential nitrogenase rates in the presented strain, more work needs to be done to clarify the nature of this feature. Deeper discussion of the published research on the two nitrogenase types, especially in R. capsulatus or others with two nitrogenase operons, would help support the proposition that there is a "fast" or "slow" nitrogenase. To facilitate this, Fig. 6 needs to be completely overhauled, as it is currently far too large with too small text and nothing can be interpreted regarding the nitrogenase operons. A simpler tree or trees showing the two Rdv. tesquicola clusters' relationship to other species is necessary to show this relationship.
We have changed Fig. 6 in order to make it more comprehensible, leaving only the Rhodobacterales members. Now the two clusters of “slow” and “fast” nifH genes are clearly detectable. The larger tree including other phyla is moved to the Supplementary Materials as Figure S1.
Unfortunately, we have found little information in the literature regarding the nitrogenase types that can be interpreted as “fast” and “slow”. We have added the discussion of the probable functioning of different nitrogenases occuring in the same organism. Besides, we are planning to further clarify the functioning of the “fast” and “slow” nitrogenase types in Rhodovulum and hope to present it in our future works.
- 1 & 2 These are redundant and both are not necessary for the main body of the manuscript. I would recommend moving one to supplemental.
We think that both Fig. 1 & 2 should be present in the main body of the manuscript to better illustrate the phylogenetic position of the investigated strains.
- 1 & 6 Please clearly highlight the data sequenced in this work. The text is also incredibly small on both figures.
The information is added to the Materials and Methods section: the genome of R. tesquicola was sequenced for this work and the genomes of other Rhodovulum members as well as the sequences of genes used in this research were obtained from NCBI Assembly database.